# Regression-Based Normative Data for Independent and Cognitively Active Spanish Older Adults: Digit Span, Letters and Numbers, Trail Making Test and Symbol Digit Modalities Test

**DOI:** 10.3390/ijerph18199958

**Published:** 2021-09-22

**Authors:** Clara Iñesta, Javier Oltra-Cucarella, Beatriz Bonete-López, Eva Calderón-Rubio, Esther Sitges-Maciá

**Affiliations:** 1SABIEX, Universidad Miguel Hernández de Elche, Av. de la Universidad, 03207 Elche, Spain; clara.inesta@goumh.umh.es (C.I.); bbonete@umh.es (B.B.-L.); eva.calderon@goumh.umh.es (E.C.-R.); esther.sitges@umh.es (E.S.-M.); 2Department of Health Psychology, Miguel Hernandez University of Elche, 03202 Elche, Spain

**Keywords:** cognitively active, cognitive impairment, neuropsychological assessment, normative data, older adults

## Abstract

In this work, we developed normative data for the neuropsychological assessment of independent and cognitively active Spanish older adults over 55 years of age. Method: Regression-based normative data were calculated from a sample of 103 non-depressed independent community-dwelling adults aged 55 or older (67% women). Raw data for Digit Span (DS), Letters and Numbers (LN), the Trail Making Test (TMT), and the Symbol Digit Modalities Test (SDMT) were regressed on age, sex, and education. The model predicting TMT-B scores also included TMT-A scores. Z-scores for the discrepancy between observed and predicted scores were used to identify low scores. The base rate of low scores for SABIEX normative data was compared to the base rate of low scores using published normative data obtained from the general population. Results: The effects of age, sex, and education varied across neuropsychological measures. Although the proportion of low scores was similar between normative datasets, there was no agreement in the identification of cognitively impaired individuals. Conclusions: Normative data obtained from the general population might not be sensitive to identify low scores in cognitively active older adults, incorrectly classifying them as cognitively normal compared to the less-active population. We provide a friendly calculator for use in neuropsychological assessment in cognitively active Spanish people aged 55 or older.

## 1. Introduction

The population aged 65 years or older is expected to rise worldwide in the coming decades. The United Nations [1] predicted an increase from 9% in 2020 to around 16% in 2050. As reported by the Eurostat database, 20% of people in Europe are aged 65 or older and this percentage is estimated to increase to 30% by 2070. According to the Spanish National Statistics Institute [2], Spain is one of the countries with the highest rate of older people in Europe, with 18.58% of people aged 65 years or older.

Since age is the main risk factor for dementia [3,4], the increase in the proportion of older people is associated with an increase in the incidence and prevalence of cognitive impairment and dementia [5,6]. The number of people living with dementia worldwide is currently estimated at 50 million, with dementia being the leading cause of disability and dependence during aging [7]. A recent meta-analysis reported a 12.4% prevalence of dementia in Europe and 5–9% in Spain in people older than 65 [8]. Previous research has found that people diagnosed with Mild Cognitive Impairment (MCI) are at a higher risk of developing dementia [9]. Thus, in the absence of effective pharmacological and non-pharmacological treatments for dementia [10,11,12], early detection of cognitive impairment during aging has become a major research topic.

Neuropsychological assessment is essential to identify pathological cognitive changes during aging [13,14]. Standardized tests are administered in order to assess the functioning of different cognitive domains such as attention, memory, language, visuospatial abilities, and executive functions. Performance is interpreted by comparing individuals’ scores with scores from a reference group [13]. As raw scores in cognitive tests are affected by demographic variables such as age, sex, or educational level [15,16,17], normative data are used to transform them into relative measures corrected for the influence of these variables [16,18] and to provide a framework in which theses scores can be located and interpreted. Thus, selecting appropriate normative datasets is necessary for accurately interpreting the results of the neuropsychological assessment, and for reducing the probability of false diagnoses of cognitive impairment [15,19].

Different approaches to developing normative data have been reported. The simplest procedure is based on the tests’ score distribution to generate norms from the means and standard deviations. This strategy can be used with the entire sample or stratifying the sample by age [20,21], sex [22], and education [20,21,23]. Means and standard deviations within each subgroup are used to transform raw scores into easily interpretable measures such as *Z* scores, T-scores, scaled scores, or percentile ranks [22]. This method has some limitations: First, it is based on a series of arbitrary strata [24], assuming which person variables are predictive of the test score; second, the estimated population means and variances can be less reliable when dividing the sample into subgroups than using the whole sample [25]. A more advanced procedure to develop normative data is using multiple linear regression models to estimate an individual’s predicted level of performance, based on sociodemographic variables such as age, sex, and education. The difference between the predicted and the observed score (residual values) is then standardized and interpreted [26,27,28]. A different procedure for clinical classification is the Receiver Operating Characteristic curve (ROC) analysis, which is used to determine the cut-off score with the optimal balance between sensitivity and specificity [29,30]. The area under the ROC curve (AUC) offers an index of the test’s overall discrimination accuracy, with values close to 1 suggesting a high diagnostic accuracy.

### 1.1. Active Aging

Although brain changes during normal aging entail changes in some cognitive abilities [31], certain activities are considered protective factors against cognitive decline, such as continued learning and engagement in socially and cognitively stimulating activities during aging [32,33]. This protective link is mostly attributed to an increased cognitive reserve, which compensates for brain changes in normal aging and delays the clinical expression of cognitive impairment despite underlying brain pathology caused by neurodegenerative processes [34,35]. Supporting these hypotheses, frequent participation in cognitive activities has been associated with slower late-life cognitive decline [36] and a reduced risk of developing MCI and dementia [37].

As a response to the challenges of population aging, the concept and policies of “Active Aging” emerged. The Active Aging Framework promotes the optimization of opportunities for health, participation, and security with the aim of improving the quality of life as people age [38,39]. This notion emphasizes the importance of an active lifestyle and the benefits of life-long learning [40,41]. From this perspective, university programs for seniors (UPS) have become an important resource for increasing opportunities for active aging, improving several aspects such as health, psychological well-being, cognitive functioning, autonomy maintenance, and social participation [41,42,43]. In recent decades, UPS have spread worldwide [40,44] and have prompted an increase in the number of older adults that undertake university courses. In Spain, according to the State Association of University Programs for Older Adults (AEPUM), the number of adults aged 55 or older enrolling in these programs increased from 23,000 during the 2005–2006 academic year to 63,173 in 2018–2019 (https://www.aepumayores.org/) (accessed on 20 June 2021).

Older people who participate in university courses live independently in their everyday life and seek continued personal development and social interactions through these educational programs [45]. It has been reported that the motivations to attend these programs are to feel active, to invest in personal development, and to gain new knowledge and social contacts [45,46]. The evidence also suggests that individuals who engage in UPS are cognitively more active than same-age people in the general population. The tendency to engage in these courses has been related to a larger number of individual and community-based active practices. Thus, cognitively active people read more frequently, do more physical exercise, attend more cultural events, and participate more in social activities [42,47].

It has been reported that cognitively stimulating activities in mid-life [48] and late life [49] contribute to cognitive reserve independently of education. Christensen et al. [50] found that the level of activity in everyday life influenced cognitive performance and accounted for a greater proportion of variance in older people’s cognitive functioning than the level of education. In line with these results, in a post-mortem study, Reed et al. [51] found that cognitive activities during adulthood have a higher influence than the level of education in determining cognitive reserve. Thus, active aging is related to a series of practices in everyday life that differs from same-age adults from the general population, contributing to a higher cognitive reserve that may preserve or enhance their cognitive function.

### 1.2. Active Aging and Neuropsychological Assessment

There is evidence suggesting that active older adults’ lifestyles will affect performance on neuropsychological assessment irrespective of years of formal education. Active older people are likely to outperform non-active individuals on cognitive tests [50,52]. Even though normative data are demographically corrected by education, they do not account for the characteristics of active aging, and therefore, they might be less sensitive for identifying cognitive impairment among cognitively active older adults with higher performance levels. To the authors’ knowledge, there are no normative data for Spanish active older adults. This implies that active older adults might present a diagnostic challenge in conditions such as cognitive impairment and AD, as pathological changes might go undetected in the neuropsychological assessment. To fill that gap, this study developed normative data on the assessment of attention, processing speed, and working memory through four cognitive tests widely used as part of the neuropsychological assessment.

The Digit span forward (DSF) and backward (DSB) [53] are two frequently used measures of attention and working memory. The Spanish edition of the WAIS-III includes normative data for Spanish individuals. There are also normative data of this test for subjects over 50 in Spain within the NEURONORMA Project [54]. Some studies with healthy controls and MCI and AD patients have reported the effectiveness of both subtests to identify subtle impairments and to detect MCI [55], and to differentiate people with MCI and AD [56]. Lortie et al. [57] found that individuals with MCI declined in performance over 6 months, suggesting that both subtests are a reliable measure for monitoring the disease progression. The backward digit span subtest has also been reported to be a key variable discriminating between dementia subtypes such as AD and Dementia with Lewy Bodies [58].

The Letters and Numbers subtest [53] is used as a measure of working memory. Some studies provided normative data for this test in Spain [54] and Latin America [59] with adults from the general population. Kessel et al. [60] found that MCI and AD patients performed worse on this subtest compared with healthy controls. Worse performance in AD compared with MCI patients was also revealed, meaning it is suitable to differentiate people with MCI and AD.

The Symbol Digit Modalities Test (SDMT) [61] is used as a measure of information processing speed. Norms obtained with healthy adults from the general population have been published in Spain [54] and in Latin America [62]. Smith also included normative data in Spanish for an age range of 18 to 85 years for two schooling groups [61]. Performance on the SDMT is a significant predictor of conversion from cognitively normal to MCI [63] and of progression from MCI to AD [64]. It is also one of the most commonly used tests in the assessment of Multiple Sclerosis [65], Huntington’s Disease [66], and Parkinson’s disease [67].

The Trail Making Test (TMT) [68] is widely used as a measure of attention and processing speed [15]. TMT normative data have been reported for adults over 50 in Spain within the NEURONORMA Project [54] and for adults aged 18–95 in Latin America [69]. The TMT is used to screen for neurodegenerative diseases in older adults, such as Alzheimer’s Disease [70], Parkinson’s Disease [71], and Huntington’s disease [72]. Both parts A and B are sensitive to the detection of both progressive cognitive impairment and dementia [19,73].

Because the likelihood of diagnostic errors among active older adults could potentially increase by using norms obtained from the general population, normative data adapted to the specific characteristics of this population are needed. Thus, the aim of this study is to provide normative data for these four commonly used neuropsychological tests with a sample of cognitively active Spanish older adults who attend university courses. Since the cognitively active population has higher cognitive performance independently of age and years of education, the hypothesis is that they will obtain a lower rate of low scores using normative data from the general population than with normative data obtained from the cognitively active older population.

## 2. Materials and Methods

### 2.1. Participants

This is a cross-sectional observational study with cognitively healthy individuals living independently in the community. Voluntary participants were recruited consecutively from the University for Seniors (SABIEX) at the Universidad Miguel Hernández de Elche (Spain) from October 2019 to July 2021. SABIEX is a comprehensive program for the promotion of active and healthy aging and includes an academic university program for people aged 55 years or older, covering topics such as economics, physiology, sociology, politics, arts, among others, as well as the possibility of participating in different activities such as seminars, voluntary work, theater workshops, and radio programs.

Inclusion criteria for participation were (a) being 55 years old or older, (b) being cognitively normal (CN) without subjective cognitive complaints, and (c) living independently in the community. Participants were classified as CN if they had (a) Mini-Mental State Examination [74] scores higher than 23, (b) Clinical Dementia Rating scale [75] scores equal to 0, and (c) Instrumental Activities of Daily Living [76] scores 7 or higher. Exclusion criteria were (a) unwillingness to participate in the neuropsychological assessment, and (b) the presence of vision and/or hearing impairments that might have impeded the administration of cognitive tests. Participants were not excluded based on a history of medical conditions (e.g., diabetes, high blood pressure, cancer, psychiatric disorders, metabolic disease) in order to assure that the sample is representative of the population of people over 54 years in Spain [77,78]. All participants were born and raised in Spain and had Spanish as their first language.

### 2.2. Procedure

Participants were invited to participate voluntarily in the neuropsychological assessment and were assessed individually by a board-certified clinical neuropsychologist (JO-C) and trained undergraduate and master’s or PhD degree students. The neuropsychological assessment was performed in one session and took approximately 90 min. Participants signed the informed consent prior to enrollment and provided personal and family health history. Personal data were coded anonymously. This project was approved by the UMH Ethics Committee (DPS.ESM.01.19).

A sociodemographic questionnaire was created for this project to collect data on gender, age, years of formal education, residence zone (rural, urban), civil status, household context, and medical history. The neuropsychological tests included in this work were administered as part of a larger neuropsychological assessment covering several cognitive domains. The tests were administered in a pre-established order so that there was no interference between different tasks (e.g., interaction between language and verbal memory tasks). The tests included in the neuropsychological assessment have been previously described [79]. We calculated normative data with more than 100 participants because using linear regression models with a sample size greater than 100 and z ≤ −1.28 gives a number of true positive and true negatives around the 95% confidence interval [80].

### 2.3. Materials

Subjective cognitive complaints and general cognitive functioning were assessed with the CDR scale and the MMSE, respectively. Depressive symptoms were assessed with the Yessavage Geriatric Depression Scale [81] (GDS).

Attention and working memory were assessed with the Digit Span Forward (DSF) and backward (DSB), and Letters and Numbers (LN) subtests from the Wechsler Adult Intelligence Scale—3rd edition [53], and the Trail Making Test Part B [68]. Speed of processing information was assessed with the TMT Part A (TMT-A) and the written version of the Symbol Digit Modalities Test [82]. These tests were administered in the order previously described.

#### 2.3.1. Digit Span Forward and Backward

In the DSF test, the examinee is requested to repeat a series of numbers in direct order. In the DSB test, the examinee is requested to repeat a series of numbers in reverse order. In this study, the outcome variables were the longest series recalled, i.e., the maximum number of digits correctly repeated (span score) without any error in one of the two trials. For the DSF, the maximum raw score is 9, and for DSB, it is 8.

#### 2.3.2. Letters and Numbers

The LN test requires a series of letters and numbers of increasing length to be repeated. The individual is read the combination of numbers and letters in random order and instructed to repeat back the numbers first, in ascending order, followed by the letters in alphabetic order. The study variable was the longest series recalled, for a maximum of 8 items.

#### 2.3.3. Trail Making Test

The TMT consists of two parts (TMT-A and TMT-B). TMT-A contains circles numbered from 1 to 25 randomly arranged on a sheet of paper. The participant is required to draw a line connecting the circles in ascending order. TMT-B contains numbers from 1 to 13 and letters from A to L. The participant is required to connect the circles alternating between numbers and letters in ascending order. Participants are instructed to complete both parts as fast as possible while maintaining accuracy. The errors were pointed out immediately by the examiner and corrected by the participant. The outcome variable was the time taken to complete the tasks in seconds.

#### 2.3.4. Symbol Digit Modalities Test

In this test, a key box with 2 rows is presented at the top of the page with 9 unique symbols associated with 9 unique symbols. One hundred and twenty symbols are then shown, each with a blank space underneath. The participant is required to consecutively fill in each blank with the number that matches each symbol as fast as possible. After 10 practice items, the participant continues the task for 90 s. This subtest was administered according to the standard procedures described in the test manual [82]. The outcome variable was the number of correct responses, and the maximum score is 110.

### 2.4. Statistical Analyses

#### 2.4.1. Calculation of Normative Data

The regression-based normative data were calculated using age, sex, and education as predictors and raw scores as outcomes for each variable. The linear regression model can be written as
(1)$$Y^{’}=\alpha +\beta_{1}*X_{1}+\beta_{2}*X_{2}+\ldots +\beta_{i}*X_{i}+\in N\left(0,1\right),$$
where *Y*’ is the predicted score, α is the intercept, $X_{i}$ is the score of variable *i*, and $\beta_{i}$ is the beta coefficient associated with variable *i*. The intercept (α) indicates the value of the response variable when all the predictors are equal to 0, whereas the beta coefficients indicate the mean change in the response variable for a 1-unit increase in the predictor while holding constant the rest of the predictors. Because age and education in our sample did not have values of 0, we first transformed both the age and education variables for the intercept to be interpretable. We transformed data on age and education for each participant, taking the lower value in the distribution as reference (see Table 1 for descriptive statistics). Thus, if the minimum age in the sample is 55, the age of a participant aged 60 was recoded as 5. These transformed variables are referred to as Age_Min_ and Education_Min_ throughout the manuscript. Sex, Age_Min_, and Education_Min_ were included as predictors in the first step in a forward multiple linear regression model. The second and third steps added the quadratic Age_Min_ and Education_Min_ and the cubic Age_Min_ and Education_Min_, respectively, so as to analyze possible curvilinear relationships. This procedure was used for predicting each of the 6 variables independently.

As some variables in the neuropsychological battery are not independent of each other, normative data calculated independently might be misleading if relevant information is not included in the model. This is the case for the TMT-B, whose scores are not independent of scores on the TMT-A. To illustrate the dependency of scores, if two individuals score 120 s on the TMT-B, this score corresponds to an average performance (e.g., z-score = 0). Individual A obtains a z-score = 1.5 on the TMT-A, whereas individual B obtains a z-score = 0 on the TMT-A. Reasonably, an average score on the TMT-B cannot be interpreted the same way for an individual with high-level performance on the TMT-A compared to an individual with average performance on the TMT-A. Although TMT-B scores are similar for both individuals, TMT-B scores for individual A are likely showing a greater decline compared to TMT-B scores for individual B. To improve the interpretation of scores, it is necessary to know how frequent it would be to show such a decline in the TMT-B according to scores on the TMT-A. For this reason, we calculated normative data for the TMT-B with a regression model including the same predictors plus TMT-A scores, which will be referred to as TMT-B_SABIEX,_ in order to differentiate these normative data from the TMT-B normative data calculated independently of the TMT-A. Since TMT-A scores did not have a value of 0, this variable was first transformed for the intercept to be interpretable by taking the lower value in the distribution as a reference. The transformed variable will be referred to as TMT-A_min_ throughout the manuscript.

#### 2.4.2. Comparing Normative Data Sets

To analyze whether normative data for cognitively active older individuals provide different data compared to normative data obtained in the general population, we compared the number of low scores shown by our sample when using either the SABIEX or the NEURONORMA normative data. The NEURONORMA normative data were developed with individuals recruited in the general population [83], and provide age-, sex-, and education-corrected Scaled Scores (SS). Unlike SABIEX normative data, which provide residual z-scores, NEURONORMA normative data provide SS to interpret performance, which limits the selection of a cut-off point to define low scores. To avoid using different scales, low scores were identified as SS equal to or lower than 6 using NEURONORMA, and as z-scores equal to or lower than −1.28 using SABIEX normative data. Individuals were labelled as showing low scores when showing at least one low score as defined above.

As the same individuals were categorized as showing low scores based on both SABIEX and NEURONORMA normative data, the McNemar test (corrected for continuity) for related proportions [84] was used to analyze whether the number of individuals with one or more low scores differed between normative datasets. Additionally, because it is important to not only know whether the proportions of individuals labeled as impaired differ, but also whether the same individuals show one or more low scores using both normative datasets, the Fleiss’ kappa [84] interrater correlation coefficient for categorical data was used to analyze the level of agreement between SABIEX and NEURONORMA normative data. According to Fleiss et al. [84], agreement beyond chance can be interpreted as poor, fair to good, and excellent for values of 0–0.40, 0.41–0.75, and >0.75, respectively.

## 3. Results

From a pool of 105 consecutive participants, two were not included because of MMSE scores <24. The sample was composed of 103 participants (69 women, 67%). Participants’ age ranged from 55 to 87 and years of education from 3 to 22 (not including University for Seniors). Descriptive statistics for demographic variables and MMSE, IADL, and GSD scores are provided in Table 1. Statistically significant differences were found between sexes in age (*p* = 0.003, 95%CI = 1.43, 6.67), with men (M = 68.47; *SD* = 6.50) being older than woman (M = 64.42; *SD* = 6.22), but not in years of education (*p* = 0.583, 95%CI = −1.04, 1.85), MMSE (*p* = 0.114, 95%CI = −1.10, 0.12), IADL (*p* = 0.485, 95%CI = −0.03, 0.06), or GDS (*p* = 0.240, 95%CI = −2.22, 0.56). Most participants (59.2%) were married and were living with another person (72.8%). A total of 63 participants (61.2%) reported a history of medical illnesses (Table 2) and 41 (39.8%) were currently taking medication. Performance on neuropsychological tests is shown in Table 3. There were no statistically significant differences between sexes in tests performance. No statistically significant differences were found in tests performance between participants with or without medical history (all *p*’s > 0.05), nor between participants who were or were not taking medication (all *p*’s > 0.05).

### 3.1. Calculation of Normative Data

The effects of age, sex, and education varied across neuropsychological measures. The multiple linear regression models are presented in Table 4. Regression analyses showed that age was significantly associated with LN, TMT A and TMT-B_SABIEX_, and SDMT. Age_Min_^2^ was significantly related to TMT-B and TMT-B_SABIEX_. Education had significant effects on DSF, TMT-A, TMT-B, TMT-B_SABIEX_, and SDMT. Education_Min_^2^ was associated with DSB, and sex had no effect on the neuropsychological tests included in this paper. Of relevance to this study was the association found in dependent tasks within the TMT. The regression analyses for the TMT-B including scores on the TMT-A (TMT-B_SABIEX_) showed that performance on the TMT-A is significantly associated with performance on the TMT-B.

For all multiple linear regression models, multicollinearity (variance inflation factor [VIF] ≤ 10) was evaluated. VIF values in all models were well below 10 and collinearity tolerance values did not exceed the value of 1 [85].

### 3.2. Comparing Normative Data Sets (NEURNORMA-SABIEX)

Using NEURONORMA normative data and taking a scaled score of 6 or lower as the cutoff for a low score, 30 participants (29.70%) had at least one or more low scores among the five measures. Using SABIEX normative data, with TMT-A and TMT-B as independent and a z-score *≤* −1.28 as the cutoff for low score, 32 participants (31.68%) had at least one low score among these measures (see Appendix A). The difference was not statistically significant (McNemar *χ*^2^ (*n* = 101) = 0.029, *p* = 0.863). The Fleiss’s Kappa coefficient showed poor agreement in the individuals labeled as showing one or more low scores using NEURONORMA and SABIEX data (k = 0.209, *p* = 0.036).

Using the SABIEX normative dataset with the TMT-B conditional on the TMT-A (TMT-B_SABIEX_), 30 participants (29.70%) showed at least one low score. The McNemar Test showed no statistically significant differences when compared to NEURONORMA (McNemar *χ*^2^ (*n* = 101) = 0.031, *p* = 0.859). Again, there was poor agreement identifying low scores between normative datasets (*k* = 0.241, *p* = 0.015).

The number of low scores shown by fewer than 10% of the sample was two or more with the three normative datasets: NEURONORMA (7.8%; SS < 6), SABIEX (8.7%; *z* ≤ −1.28), and SABIEX taking the TMT-B as conditional on the TMT-A (9.9%; *z* ≤ −1.28).

MMSE scores were compared between individuals with and without low scores within each normative dataset. Using SABIEX normative data, statistically significant differences were found on the MMSE scores (*p* = 0.010, 95%CI = 0.22, 1.62) between individuals showing one or more low scores (M = 28.75; *SD* = 1.40) and those showing no low scores (M = 27.83; *SD* = 1.64). Using NEURONORMA normative data, there were no statistically significant differences in the MMSE scores (*p* = 0.798, 95%CI = −0.62, 0.80) between individuals with one or more low scores (M = 28.50; *SD* = 1.34) and those with no low scores (M = 28.59; *SD* = 1.50).

### 3.3. Comparing Trail Making Test (NEURONORMA-SABIEX)

Separated contrast analysis was performed to compare the proportion of low scores when using the TMT-B independent of the TMT-A (TMT-B_NEURONORMA_) and the TMT-B conditional on the TMT-A (TMT-B_SABIEX_).

Using TMT-B_NEURONORMA_, the percentage of low scores was 12.75%, and with TMT-B_SABIEX_, 4.9%. The corrected McNemar test was not statistically significant (McNemar χ^2^ (*n* = 102) = 2.722, *p* = 0.099), probably because none of the individuals showed a low score with both normative datasets. Interestingly, there was no agreement between the two normative datasets (k = −0.097, *p* = 0.344) when classifying individuals as showing low scores.

A friendly calculator of z-scores for DSF, DSB, LN, SDMT, and TMT is available for clinicians and researchers at https://drive.google.com/file/d/1p-RDT6F85EsXPxALV-l6R8Sq-E4qGEr0/view?usp=sharing.

## 4. Discussion

This work aimed to provide regression-based normative data for the Digits, Letters, and Numbers, TMT, and SDMT tests for cognitively active Spanish adults aged 55 or older. Additionally, and compared with other normative studies [69,86], our study introduced a novel approach for the calculation of normative data for the TMT, that is, considering the dependency of related tasks by calculating normative data for the TMT-B controlling for scores on the TMT-A.

Regarding the use of the SABIEX normative data compared to normative data obtained from the general population, our results showed poor agreement in identifying low scores. Considering this discrepancy in the classification and that normative data must be adapted to specific characteristics of individuals for adequate score interpretation [15,19], our data suggest that using normative data obtained in the general population for the neuropsychological assessment of active older adults might be associated with an increase in the number of misdiagnoses by erroneously identifying low scores.

Regarding the results in tests composed of related measures, other studies have shown that for an accurate interpretation of performance in the assessment, it is necessary to consider the correlation among different measures [87,88]. Even though a moderate correlation (r = 0.31–0.6) between Trail A and B has been reported [15], normative data for the TMT are usually calculated treating both parts as independents. This work shows that when TMT-B is analyzed considering scores in trail A (TMT-B_SABIEX_), the results are different from those obtained when they are considered independent. In our study, a strong correlation between trail A and B (r = 0.62) was found, as well as a significant contribution of TMT-A scores in the prediction model for TMT-B. This finding supports that interpreting them as independent might increase the likelihood of diagnostic errors in the identification of cognitive impairment. This conclusion is supported by the data indicating that all the participants classified as showing a low score on the TMT-B using the NEURONORMA dataset are classified as showing average scores on the TMT-B conditional on TMT-A scores (TMT-B_SABIEX_).

The demographic variables included in the prediction models (age, sex, and education) had different effects across the neuropsychological tests studied in this work. As in previous works [27,89], our study included quadratic age and education in the regression models, which allowed us to explore possible non-linear associations between these variables and performance in the tests. Overall, older age and lower education were associated with worse performance. These findings are in line with previous research showing age-related decrease in performance and positives effects of education on cognitive function [17,90]. Consistent with the results reported by Salthouse [91], the cubic term did not provide additional information over the model including the quadratic term, which showed that performance worsened for the oldest ages and increased for the highest years of education.

### 4.1. Digit Span Forward and Backwards

A relationship between older age and worse performance, as well as a positive effect of level of education, has been frequently reported [54,92]. In terms of education, in line with previous studies, our work confirms the existence of a significant effect of education on both DSF and DSB [54,93]. However, the relationship between education and DSB showed a curvilinear pattern, and compared to other studies, the effect size [93] was small for both tests (r^2^ = 0.05). Contrary to the previous studies [93,94], we did not find any effects of age on the digit span.

The fact that our results differ from previous studies with older adults in terms of age and the effect size of education is of special relevance. Several works have reported a decrease in performance on the DS beyond 65 years of age and have also found that performance on the DS is influenced by level of education [19,93,94]. However, it has been reported that frequent cognitive activity is associated with a reduced rate of cognitive decline in older adults [95]. One possible interpretation is that the influence of age on cognitively active adults’ performance might not follow the same pattern than in the general population and, therefore, specific normative data for these populations are needed. The small effect size for education suggests that cognitive activities during adulthood have a higher influence than the level of education in determining cognitive reserve [51].

### 4.2. Letters and Numbers (LN)

Regarding the effects of age on performance, our results show a negative linear relationship between age and LN performance, with a lack of contribution of quadratic age. These findings disagree with a previous study that reported a significant curvilinear decrease with age [96] with a sample of young and older adults. Differences in the age range of the sample, from 55 to 87 in our study and 18 to 89 in Myerson et al. [96], may be responsible for the variability. However, consistent with our results, in the study of Myerson et al. [96], a pronounced linear decline with age is found in individuals beyond 60 compared to the 20- to 60-year-olds.

### 4.3. Trail Making Test

In line with previous studies, our results indicate that both parts A and B are associated with age and educational level but not with sex [54,97,98]. Moreover, even in previous works reporting statistically significant associations between sex and TMT [86] or between sex and TMT-B [99], these associations were small, with sex accounting for a negligible proportion of the variance on the TMT (<1%). Some studies reported a linear relationship between age and completion time in both parts [86,99]. In our study, TMT-B scores are affected by the quadratic term of age, suggesting that performance on the TMT-B worsens more markedly in the oldest population. Besides, our results support that parts A and B should not be treated as independent when interpreting performance so as to decrease the likelihood of false positive diagnoses. The TMT-B_SABIEX_ normative data might provide clinically useful data as a complement to existing general population-based norms for evaluating Spanish older adults. Future works should be conducted on clinical settings to examine whether the use of both normative datasets adds different and valuable information in the assessment of attention and processing speed.

### 4.4. Symbol Digit Modalities Test

In line with previous works, we found that younger age and higher education were significantly associated with better performance on the SDMT [54,62,89,100,101]. Contrary to Ryan [101] and Kiely [100], but in line with Peña-Casanova [54] and Arango-Lasprilla [62], we did not find a significant effect of sex.

Since older people have an increased risk of cognitive impairment [102], the availability of appropriate and reliable normative data is essential for early and accurate identification of pathological changes in cognition. Although several studies have reported on normative data for older adults, these studies used general population-based samples. The finding that there is no agreement in the individuals labeled as showing low scores when using normative data obtained from the general population and SABIEX-specific normative data highlights the need for specific normative data for highly cognitively active people.

An important aspect that distinguishes these normative data is that we did not exclude participants with a history of medical conditions that could affect neuropsychological functioning. It has been suggested that less rigorous exclusion criteria might decrease the sensitivity of the normative data to identify true cognitive impairment [103]. However, since both the incidence and prevalence of chronic diseases [104] and multimorbidity [105] increase with age, the presence of medical conditions is frequent in the older population. Therefore, including only healthy older adults in a normative study could bias the results as the sample would be unrealistic and not representative of the population that will be assessed with these normative data. Moreover, using an extremely healthy sample might increase the risk of overdiagnosing cognitive impairment in clinical settings. As in other normative studies for older adults [26,83], we ensured the normal cognitive functioning of participants included in our sample through the scores in the MMSE, CDR, and independence in the ADLs.

Regarding the clinical applicability of these norms, the neuropsychological tests described in this paper might be useful in the process of diagnosing cognitive impairment and dementia. Since active older adults may have a better cognitive functioning compared to same-age people from the general population [50,95], impaired performance could be more difficult to identify through tests standardized on the general population. The normative data reported in the present work might be especially helpful for clinicians and researchers to accurately interpret scores of older adults who continue to lead a very active life during aging, identifying lower-than-average performance more accurately and, thus, reducing the risk of diagnostic errors if low scores are to be used to diagnose cognitive impairment. Since individuals with MCI are at greater risk of AD [9,106], the SABIEX normative data might help to identify MCI with greater certainty in highly cognitively active Spanish individuals. One of the strengths of this work is that we provide normative data for five neuropsychological measures, which allows the comparison of an individual‘s performance across the different normed tests.

### 4.5. Limitations

These normative data should be interpreted with limitations. First, the sample used to calculate them was taken from university courses for seniors. This restricts the generalizability to the population with this characteristic. In Spain, during the 2019–2020 academic year, the number of adults that participated in these courses amounts to 35.199 (https://www.aepum.es/) (accessed on 20 June 2021), which represents 0.22% of the population aged 55 or older (https://www.ine.es) (accessed on 13 February 20221). However, different studies have reported that other active practices during aging, such as volunteering, contribute to the maintenance of cognitive reserve and positively impact older adults’ cognitive function [107,108]; therefore, it is recommended to analyze whether these normative data are also appropriate to properly interpret the performance of older adults engaged in cognitively stimulating activities other than university courses.

Another potential limitation of the present study is that, due to the composition of the sample, these normative data will be useful only for people between 55 and 87 years old and with 3 to 22 years of education. Additionally, taking into account the well-documented effects of culture on the discrepancy in performance in the different cognitive domains [109,110], another limitation of these normative data is that they are only applicable to the Spanish population. Their use with other Spanish-speaking populations, with different cultural backgrounds, is limited. Some studies have found differences in performance in the tests included in this work among individuals from different Spanish-speaking countries [62,69], and using these normative data might result in diagnostic errors.

A further limitation of this study is that the normative data are obtained with cognitively active adults, but they have not been applied in clinical populations. It is still unknown if they are adequate to identify cognitive impairment. Future studies should be conducted with clinical populations to help to clarify the clinical usefulness of these normative data. Therefore, we suggest using these normative data as a supplement of existing general-population-based norms until their clinical utility is analyzed. Another limitation is the fact that z-scores equal to or lower than −1.28 were used to interpret performance and define low scores, whilst the most commonly used cut-off point to interpret cognitive impairment is at least 1.5 standard deviations below the mean.

Lastly, since aging is associated with changes in the brain structure and in the functional connectivity related to cognitive processes [111,112], a limitation of the present study is the lack of neuroimage profiles to analyze correlates between brain structure, functional connectivity, and participants’ variability in cognitive functioning (e.g., whether the number of low scores is associated with different structural brain alterations or the connectivity between different brain areas). By analyzing our results together with magnetic resonance images (MRI), a more complete understanding of the effects of aging on network function, brain structure, and cognitive function would be obtained. Future works are warranted to identify the association of a cognitively active lifestyle and cognitive function, brain structure, and brain connectivity.

## 5. Conclusions

In conclusion, the present work provides normative data for a cognitively active Spanish population that may help to identify cognitive impairment during aging, improve diagnostic precision, and reduce diagnostic errors. Our findings highlight the importance of using appropriate normative data, relevant to the population being assessed. Despite the availability of Spanish normative data for the older population, our results suggest that the accuracy in the interpretation of active older adults’ performance might be maximized using population-specific normative data.

## Figures and Tables

**Table 1 ijerph-18-09958-t001:** Demographic statistics and performance on MMSE, IADL, and GDS.

Variable	M	SD	Range
Age	65.76	6.567	55–87
Education	11.44	3.463	3–22
MMSE	28.48	1.481	25–30
IADL	7.99	0.099	7–8
GDS	4.32	3.355	0–14

M: mean, SD: standard deviation.

**Table 2 ijerph-18-09958-t002:** Number (and %) of participants with medical history (*n* = 103).

	*n*	%
Anxiety	19	18.4
Depression	8	7.8
Epilepsy	2	1.9
Stroke	4	3.9
CVD	12	11.7
Hypo/hipertyroidism	7	6.8
Cancer	9	8.7
DM	4	3.9
HBP	13	12.6
TBI	3	2.9
COPD	1	1.0
RA	2	1.9
Others	14	11.7

CVD: Cardiovascular disease, DM: Diabetes mellitus, HBP: High blood pressure, TBI: Traumatic brain injury, COPD: Chronic obstructive pulmonary disease, RA: Rheumatoid arthritis, Others: Optic nerve sheath meningioma, hypercholesterolemia, osteopenia, Chronic venous insufficiency, Dyslipidemia, dyspepsia, Cholecystectomy, Autoimmune hypoglycemia, COVID-19, Asthma.

**Table 3 ijerph-18-09958-t003:** Raw scores on neuropsychological tests.

Neuropsychological Measures	*M*	*SD*	Range
DSF (*n* = 103)	5.36	1.153	3–9
DSB (*n* = 103)	3.94	0.873	2–7
LN (*n* = 102)	4.50	1.115	2–8
TMT-A (*n* = 103)	47.79	16.249	24–140
TMT-B (*n* = 102)	119.08	57.807	41–345
TMT-B_SABIEX_ (*n* = 102)	119.08	57.807	41–345
SDMT (*n* = 102)	37.19	9.571	16–56

M: Mean, SD: Standard deviation, DSF: Digit Span Forward, DSB: Digit Span Backwards, LN: Letters and Numbers, TMT: Trail Making Test, SDMT: Symbol Digit Modalities Test.

**Table 4 ijerph-18-09958-t004:** Multiple linear regression models.

		Β	*SE*(β)	95% CI	*p* _coeff_	R^2^_Adjusted_
DSF	Intercept	4.719	0.294	4.14–5.30	<0.001	
	Education_Min_	0.076	0.032	0.01–0.14	0.021	0.043
DSB	Intercept	3.695	0.135	3.43–3.96	<0.001	
	Education_Min_^2^	0.003	0.001	0.00–0.00	0.022	0.042
LN	Intercept	5.014	0.204	4.61–5.42	<0.001	
	Age_Min_	−0.048	0.16	−0.08–−0.02	0.004	0.071
TMT-A	Intercept	48.676	4.55	39.66–57.70	<0.001	
	Age_Min_	0.741	0.232	0.28–1.20	0.003	
	Education_Min_	−1.051	0.440	−1.92–−0.18	0.019	0.111
TMT-B	Intercept	155.352	14.001	127.57–183.13	<0.001	
	Education_Min_	−6.029	1.523	−9.05–−3.00	<0.001	
	Age_Min_^2^	0.095	0.028	0.04–0.15	0.001	0.177
TMT-B_SABIEX_	Intercept	122.836	17.487	88.13–157.54	<0.001	
	TMT-A_Min_	2.043	0.284	1.48–2.60	<0.001	
	Education_Min_	−4.303	1.272	−6.83–−1.78	0.001	
	Age_Min_^2^	0.191	0.067	0.06–0.32	0.005	
	Age_Min_	−4.211	1.930	−8.04–0.38	0.032	0.456
SDMT	Intercept	36.508	2.435	31.68–41.34	<0.001	
	Age_Min_	−0.655	0.124	−0.90–−0.41	<0.001	
	Education_Min_	0.913	0.235	0.45–1.38	<0.001	0.273

DSF: Digit Span Forward, DSB: Digit Span Backward, LN: Letters and Numbers, TMT: Trail Making Test, SDMT: Symbol Digit Modalities Test, β: Regression coefficient, *SE*(β): Standard error of β, SEM: Standard error of the measurement.

## Data Availability

Data are available at request from the corresponding author.

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
