# Peer review of "Regression-Based Normative Data for Independent and Cognitively Active Spanish Older Adults: Digit Span, Letters and Numbers, Trail Making Test and Symbol Digit Modalities Test"

_ijerph, 2021, doi:10.3390/ijerph18199958_

Round 1

Reviewer 1 Report

The study provided normative data for neuropsychological tests used with cognitively active older adults. The rationale of the investigation was sound. However, the study, although, used cognitively active participants, conclusions drawn without comparing against a cognitively inactive group, can not be fully accepted. We do not know if the cognitively inactive group would have preformed anything different on those neuropsychological tests. Moreover, similar to the norms that are typically reported in the literature, the study also reported norms based on the education level. The same education based norm was criticised in their introduction by the authors.

Dependency based understanding of neuropsychological scores is a good way to interprete overall performance. In the present study, the authors have looked at the dependency of the TMT-B on TMT-A and indeed they find some dependency. I wonder if these dependencies should be explored in a wider context (consider all the tests being dependent on each other).

I am unclear about their assertion of increased sensitivity of their dependency based scores in identifying caseness. How do we know whether the typical normative data are worse in identifying cognitive problems in cognitively active older adults. What was the gold standard against these two methods, dependency and no dependency based norms, were evaluated to make that assertion of increased sensitivity of the dependency based norms?

Reviewer 2 Report

Authors developed normative data for neuropsychological assessment of independent and cognitively active Spanish older adults over 55 years of age using regression-based normative data from a sample of 103 non-depressed independent community-dwelling, and provided a calculator in neuropsychological assessment in cognitively active Spanish people aged 55 or older.

Below are my comments:

  1. Title. “SABIEX Regression-Based Normative Data for Independent and Cognitively Active Spanish Older Adults: Digit Span, Letters and Numbers, Trail Making Test and Symbol Digit Modalities Test” seems misleading. “SABIEX” in the title is a bit misleading and could be removed.
  2. Abstract. The conclusion “normative data obtained from the general population might not be sensitive to identify low scores in cognitively active older adults, incorrectly classifying them as cognitively normal compared to less active population” looks not supported by the data presented in the Abstract.
  3. Materials and Methods. It is unclear what the study design is.
  4. Statistical analyses

(1) The expression (N,delta^2) in equation (1) on page 6 is not a conventional expression of standard normal distribution.

(2) To achieve zero values for age and education, the conventional method is to centralise them by subtracting their means. I would suggest that authors perform a sensitivity analysis using this method and see if there are any differences in the results between two transformation methods.

(3) It is unclear what statistical criterion was used to assess and identify possible curvilinear relationships.

(4) “the McNemar test (corrected for continuity) for related proportions [99] was used to analyze whether the number of individuals with one or more low scores differed between normative data sets”. The McNemar test is a univariate analysis method without controlling for possible confounding factors.  Why multivariate method was not used to control for possible confounding factors?

  1. Results

(1) t-values and p-values are presented. It would be preferable to present 95% confidence interval and p-values.

(2) Similarly, Chi-square values and p-values are presented. It would be more interesting to see the difference in terms of direction, magnitude and precision.

  1. Conclusion

Please add the discussions of selection bias and confounding effects in the limitations of this study.

  1. References

There are 131 references in the article, which seem too many to me.

  1. Table 4.

(1) Too many statistics are presented here. For such a linear regression analysis, I think regression coefficient, its 95% confidence interval and p-value, adjusted R squared should be enough.

(2) “(Intercept)”=>“Intercept”

(3) If p=0.000, please write as “<0.001”

  1. This is an observational study and STROBE guidelines should be followed.

Round 2

Reviewer 2 Report

Most of my comments have been addressed but the following two comments need addressing.

  1. Design of the study.  Authors describes the study as an observational study but please be specific about the study design: is it a cohort or cross-sectional study?
  2. t-values have been removed from the tables but not from the text. Please remove.
